# Synthesis and Characterisation of Self-Cleaning TiO_2_/PES Mixed Matrix Membranes in the Removal of Humic Acid

**DOI:** 10.3390/membranes13040373

**Published:** 2023-03-24

**Authors:** Yan Kee Poon, Siti Kartini Enche Ab Rahim, Qi Hwa Ng, Peng Yong Hoo, Nur Yasmin Abdullah, Amira Nasib, Norazharuddin Shah Abdullah

**Affiliations:** 1Faculty of Chemical Engineering & Technology, Universiti Malaysia Perlis, Arau 02600, Perlis, Malaysia; 2Centre of Excellence for Frontier Materials Research (CFMR), Universiti Malaysia Perlis, Arau 02600, Perlis, Malaysia; 3School of Materials and Mineral Resources Engineering, Engineering Campus, Universiti Sains Malaysia, Nibong Tebal 14300, Penang, Malaysia

**Keywords:** photocatalytic, membrane fouling, antifouling, self-cleaning membrane

## Abstract

Membrane application is widespread in water filtration to remove natural organic matter (NOM), especially humic acid. However, there is a significant concern in membrane filtration, which is fouling, which will cause a reduction in the membrane life span, a high energy requirement, and a loss in product quality. Therefore, the effect of a TiO_2_/PES mixed matrix membrane on different concentrations of TiO_2_ photocatalyst and different durations of UV irradiation was studied in removing humic acid to determine the anti-fouling and self-cleaning effects. The TiO_2_ photocatalyst and TiO_2_/PES mixed matrix membrane synthesised were characterised using attenuated total reflection-Fourier transform infrared (ATR-FTIR) spectroscopy, X-ray powder diffraction (XRD), scanning electron microscope (SEM), contact angle, and porosity. The performances of TiO_2_/PES membranes of 0 wt.%, 1 wt.%, 3 wt.%, and 5 wt.% were evaluated via a cross-flow filtration system regarding anti-fouling and self-cleaning effects. After that, all the membranes were irradiated under UV for either 2, 10, or 20 min. A TiO_2_/PES mixed matrix membrane of 3 wt.% was proved to have the best anti-fouling and self-cleaning effect with improved hydrophilicity. The optimum duration for UV irradiation of the TiO_2_/PES mixed matrix membrane was 20 min. Furthermore, the fouling behaviour of mixed matrix membranes was fitted to the intermediate blocking model. Adding TiO_2_ photocatalyst into the PES membrane enhanced the anti-fouling and self-cleaning properties.

## 1. Introduction

Fresh and clean water resources are vital for daily human activity and social development. However, the dramatic growth of population and industry causes severe pollution, resulting in a shortage of clean water in certain places [1]. Humic acid is a natural organic matter from the soil and a primary contaminant in water [2]. Humic acid causes water to become yellowish or brownish, react with disinfectants and form disinfectant by-products and the intensification of microbial re-growth in water distribution networks and destruction of microbiological, physical (unpleasant taste and odor) and chemical quality of water [3,4,5].

There are a few methods for removing humid acid: coagulation, adsorption, membrane filtration, the advanced oxidation process, and biological degradation [6]. A combined approach, a photocatalytic membrane, has a high efficiency in removing humic acid. However, a significant problem in membrane filtration is membrane fouling [7,8,9]. Therefore, a photocatalyst is chosen to be added to the membrane system. Membrane coupling photocatalytic processes can be divided into two main groups: (i) systems with the photocatalyst in suspension (slurry conditions) and (ii) systems with the photocatalyst immobilised on the membrane surface and within its pores (mixed matrix membrane) [10]. However, studies by Espindola et al. [11] show that the immobilisation of the TiO_2_-P25 minimised flux decline on the membrane due to pollutant oxidation on the surface and within the pores when compared to experiments on photocatalyst slurry dosage in a photocatalytic membrane reactor system. Other advantages are that to minimise photocatalyst loss, the post-treatment process for photocatalyst separation can be removed, and these photocatalysts will act as a barrier for more significant microorganisms and molecules, such as bacteria or organic matter [12]. When the photocatalyst is immobilised in the membrane, it provides a self-cleaning effect that will expand the membrane’s life. Among the photocatalysts used, TiO_2_ possesses excellent physical and chemical properties, including being highly hydrophilic, having anti-fouling abilities, and having photocatalytic activity [13].

TiO_2_ can degrade organic compounds into harmless substances by absorbing proper light energy. When TiO_2_ is irradiated with UV light, that has the same or more than the band gap energy. The electrons from the filled valence band (VB) are promoted to the empty conduction band (CB). Therefore, electron-hole (e^−^/h^+^) pairs are formed. The electron-hole pairs will move and undergo a redox reaction. The h^+^ reacts with hydroxide ions and water to form hydroxyl radicals. Meanwhile, the e^−^ reacts with oxygen, forming superoxide radical anions. Then, later, it will degrade foulants and perform a self-cleaning ability. In a mixed-matrix membrane, the photocatalytic activity is affected by the photocatalyst dosage. A high concentration of photocatalyst will encourage agglomeration and void formation, which will decrease the membrane’s performance. However, a low concentration of photocatalyst in the membrane delivers limited sites for the photodegradation reaction [14]. Therefore, it is important to have the correct amount of photocatalyst embedded in the membrane.

Determining membrane fouling can be performed by studying its fouling behaviour. Four membrane fouling models exist, including standard blocking, cake filtration, intermediate blocking, and complete blocking. A total blockade is assumed to mean that the pore entrances have been sealed off and the flow has been prevented, while intermediate blocking is about the same as complete blocking, but the pores are assumed to be sealed off partly and the rest of the particles will accumulate on top of other deposited particles. Cake filtration happens when the surface of a membrane is accumulated with particles, which will slowly increase in thickness and raise the resistance to flow. For standard blocking, the wall of straight cylindrical pores inside the membrane is deposited with particles. The membrane’s pores become constricted, and the membrane’s permeability reduces when the particles are deposited [15].

In this study, mixed matrix membranes were fabricated using polyethersulfone (PES) with TiO_2_ as a photocatalyst. The membrane performances were evaluated, and photocatalytic studies were performed on the assessed self-cleaning membrane under UV irradiation. The best performance membrane data were tested in a single fouling mechanism model.

## 2. Experimental

### 2.1. Chemicals and Materials

All chemicals were used as obtained, without further modification. To synthesise the membrane, the Ultrason E6020P PES polymer with a MW of 58 kDa was supplied by BASF (Ludwigshafen, Germany), while the solvent used to dissolve the polymer, N-methyl- 2-pyrrolidone (NMP), was purchased from Merck (Darmstadt, Hessen, Germany). In this study, humic acid (HA) was obtained from Sigma-Aldrich (St. Louis, MO, USA) and was used as the organic foulant. Hydrochloric acid (HCl) (1 M) and 1 M of sodium hydroxide (NaOH) supplied by Merck (Darmstadt, Hessen, Germany) were prepared for the HA pH adjustment. Titanium dioxide (TiO_2_) was used as a photocatalyst provided by Sigma Aldrich with a purity of 99.5%.

### 2.2. Synthesis of PES Membrane

The PES polymer was placed in the oven at 40 °C to remove moisture. After that, the dried PES polymer was added to N-methyl-2-pyrrolidone (NMP) solvent in a beaker containing the immersion of the sensor tip of the thermometer. The mixture was sealed instantly with parafilm before being subjected to heating to 70 °C for 60 min, followed by 4 h at room temperature at a constant stirring rate of 700 rpm. A PES of 13 wt.% was used for each run. A 200 μm thickness of the mixture was cast onto a flat glass plate and dipped into a distilled water coagulation bath for 24 h. This process accomplishes the wet phase-inversion process and removes the residual solvent. The synthesised membrane was kept in a plastic bag for further processing.

### 2.3. Synthesis of TiO_2_/PES Mixed Matrix Membrane

The pure TiO_2_ in different concentrations (1, 3, and 5 wt.%) was added to the mixture under intensive sonication and formed the TiO_2_/PES solution. A 200 μm-thick TiO_2_/PES solution was cast onto a flat glass plate. To complete the phase-inversion process, which is to induce further wet phase inversion and the residual solvent removal, the flat glass plate with casted TiO_2_/PES solution was dipped immediately into a water bath for 24 h. The synthesised TiO_2_/PES mixed matrix membrane was kept in a zipper bag for further processing.

### 2.4. Characterization of TiO_2_ and TiO_2_/PES Mixed Matrix Membranes

#### 2.4.1. ATR-FTIR Analysis

Attenuated total reflection-Fourier transform infrared spectroscopy (ATR-FTIR) was performed to determine the functional groups on the membrane surface. Fourier transform infrared spectroscopy with attenuated total reflection, ATR-FTIR (PerkinElmer Inc., Waltham, MA, USA), was employed for this purpose. All samples were scanned over the wavenumber range from 500 to 4000 cm^−1^.

#### 2.4.2. XRD Analysis

The diffraction pattern of TiO_2_ photocatalyst was investigated by an X-ray diffractometer (XRD) (Bruker D2 Phaser, Billerica, MA, USA), equipped with a copper (Cu) X-ray tube at 10mA and 30 kV.) equipped with monochromatic Cu Ka radiation (l = 0.154 nm) and operated at 40 mA and 40 kV from 5° to 80°.

#### 2.4.3. Membrane Morphology Observation

A scanning electron microscope (Hitachi TM3000, Tokyo, Japan) examined the morphology of the surface of the membrane. Image magnifications were 3000× for surface views. All specimens were freeze-dried and coated with a thin layer of platinum before observation.

#### 2.4.4. Contact Angle Analysis

The surface hydrophilicity of the membrane was characterised by the contact angle. The contact angle was determined using a contact angle meter based on the sessile droplet method (Rame-Hart, USA). The membrane was cut into small pieces. A live picture of a probe liquid dropped on the membrane surface was captured using a charge-coupled device (CCD) camera, and the software detected the contact angle directly during the measurement process. This measurement of contact angle was carried out at 25 °C. The liquid droplet was dripped on different membrane sites to get accurate results.

#### 2.4.5. Porosity Calculation

The porosity of the membrane was calculated using Equation (1):(1)$$P=\frac{V_{a}-V_{e}}{V_{a}}\times 100\%$$
where *P* is the membrane porosity, *V_a_* is the apparent volume, and *V_e_* is the existence volume.

### 2.5. Preparation of Feed Solution (Humic Acid)

To prepare a 20 ppm feed solution, four pellets of NaOH were dissolved in 60 mL of deionised water. The deionised water was topped up to 800 mL, with 0.04 g of humic acid. The solution was stirred with a magnetic stirrer until the humic acid dissolved. The pH of the solution was adjusted to 8.5 using NaOH and HCl. The solution was poured into a 2 L volumetric flask, and deionised water was added until the calibration mark.

### 2.6. Determination of Humic Acid Concentration

A calibration curve was obtained using different concentrations of HA. The absorbance of HA was determined at the end of cross-flow filtration via a UV-vis spectrophotometer (UV 1800 Shimadzu, Kyoto, Japan) at a wavelength of 254 nm. The calibration curve of absorbance against HA was plotted, and the humic acid concentration was determined.

### 2.7. Membrane Performance Evaluation

The performance of the membrane was tested by using a cross-flow filtration system that have been used in previous study [16]. Figure 1 shows the schematic diagram of the cross-flow filtration system. The effective membrane area was 4.1 cm^2^, and the filtration experiment was carried out in an in-house-fabricated cross-flow filtration cell. This filtration process comprised an electronic balance with a data acquisition system to measure the accumulated permeate mass and membrane crossflow filtration cell. The pressure in this process was maintained at 10 psi, generated by a booster pump, and controlled by a needle valve. The flow rate was maintained at 250 mL min^−1^. The pure water flux (J) was measured as a function of time until a quasi-steady flux was reached for 30 min at 10 psi. The feed solution was substituted with an HA solution with 20 mg L^−1^ and conducted under 10 psi. The accumulated permeate mass was measured using a computer-recorded electronic balance. The accumulated permeate mass was used to calculate the filtration flux (J) for up to two hours of filtration time.

After achieving a steady flux, the membrane was rinsed with pure water to remove loosely bound humic acid. The purified water flux was recorded. After that, the membrane was put under UV irradiation in a black box containing a pen-ray photochemical quartz lamp (90-0049-06 UVP, Jena, Germany) with a UV light source at a wavelength of 254 nm. The membrane was rinsed with pure water using a cross-flow filtration unit to remove the UV-degraded humic acid. The purified water flux was recorded. The flux recovery ratio (FRR) was calculated based on Equation (2).
(2)$$FRR=\frac{J_{w1}}{J_{w0}}\times 100\%$$
where FRR is the flux recovery ratio (%), J_w1_ is the pure water flux at any predetermined time (L/m^2^ h) and cleaning condition, and J_w0_ is the initial pure water flux (L/m^2^ h).

## 3. Results and Discussions

### 3.1. XRD Analysis of TiO_2_ Photocatalyst

Figure 2a,b are the XRD databases for the anatase and rutile phases of the TiO_2_ photocatalyst, while Figure 2c is the XRD experimental pattern of the TiO_2_ photocatalyst. By comparing Figure 2a,c, the dominant peak at 2θ of 25.24°, 25.29°, 36.89°, 37.74°, 47.98°, 53.89°, 55°, 62.07°, 62.67°, 68.82°, 70.12°, 70.31°, and 75.05° was referred to the anatase phase of TiO_2_ photocatalyst. In addition, the rutile phase of TiO_2_ photocatalyst was indicated by the dominant peaks at 2θ of 27.37°, 36.05°, 41.17°, and 54.26° by comparing Figure 2b,c. The TiO_2_ photocatalyst comprised 83.7% of the anatase phase, while the rutile phase was only 16.3%. Such an observation showed that the TiO_2_ photocatalyst was suitable to be added to a self-cleaning membrane because the anatase phase of TiO_2_ showed higher photocatalytic activity than the rutile phase of TiO_2_.

### 3.2. Characterisation of TiO_2_/PES Mixed Matrix Membrane

#### 3.2.1. Functionalisation Determination of Neat and TiO_2_/PES Membranes

Figure 3a is the ATR-FTIR spectrum for the TiO_2_ photocatalyst, Figure 3b is for the neat PES membrane, and Figure 3c is the ATR-FTIR spectrum for the TiO_2_/PES mixed matrix membrane. Numerous peaks in wavelength were observed in the spectrum of the neat PES membrane. The peak wavelength of 3436 cm^−1^ was due to the presence of the -OH functional group [17]. In addition, the peak at 1580 cm^−1^ is assigned to the C_6_H_6_ ring stretch in PES polymer [18]. The typical aromatic C=C band was 1490 cm^−1^ [17]. Furthermore, the wavelength peaks at 1400 cm^−1^ and 1250 cm^−1^ corresponded to the bending of C-H [17]. Chakraborty et al. [17] reported that the wavelength peak of 1325 cm^−1^ was due to the S=O bond. Homaeigohara et al. [19] reflected that the peak at 1300 cm^−1^ belonged to the asymmetrical vibrations of the sulfone group. Moreover, the characteristic band at 1230 cm^−1^ was related to a C-O-C vibration stretching bond in PES polymer. The absorption band at 1150 cm^−1^ is attributed to the symmetrical vibrations of the sulfone group based on the research of Homaeigohara et al. [19]. The peak at a wavelength of 1110 cm^−1^ was due to the asymmetric stretching of C-O [18]. In addition, the wavelength peaks at 1090 cm^−1^ and 1010 cm^−1^ corresponded to the C–C stretching of the polymer backbone [17]. The aromatic ring vibration was found at 795 cm^−1^ [18], while the peaks at 625 cm^−1^ and 695 cm^−1^ were assigned to the C-Cl bond [17]. The presence of the benzene ring, ether bond, and sulfone group confirmed the chemical structure of PES.

Figure 3c shows the ATR-FTIR spectrum of the TiO_2_/PES mixed matrix membrane. By comparing the spectra of the neat PES membrane and the TiO_2_/PES mixed membrane, the only difference found was that the peak at 3436 cm^−1^ had a greater intensity for the TiO_2_/PES mixed matrix membrane. The peak of 3436 cm^−1^ was attributed to the -OH group. This discovery was due to the addition of a TiO_2_ photocatalyst, which made the membrane more hydrophilic and improved the membrane’s anti-fouling properties [20].

There was no apparent difference between the ATR-FTIR spectra of the neat PES membrane and the TiO_2_/PES mixed matrix membrane. Such similarity may be due to the overlap of the TiO_2_ band with the PES polymer band [17].

#### 3.2.2. Morphology Surface of the Membrane

Figure 4a shows the top plan view of the SEM image of a neat PES membrane, while Figure 4b demonstrates the top plan view of the SEM image of the TiO_2_/PES mixed matrix membrane.

Figure 4a indicated that the neat PES membrane had a highly porous surface. The TiO_2_/PES mixed matrix membrane also has a highly porous membrane surface with numerous white spots (see Figure 4b). Some small cracks were observed on the membrane surface, which formed during the drying process of the membrane.

Figure 5a is the EDX spectrum for a neat PES membrane, while Figure 5b is the EDX spectrum for a TiO_2_/PES mixed matrix membrane. Figure 5a shows the EDX spectrum of the neat PES membrane. The characteristic peaks at C, O, and S belonged to the PES polymer. For Figure 5b, a new characteristic peak existed for Ti, indicating that titanium (Ti) elements were present in the mixed matrix membrane.

#### 3.2.3. Hydrophilicity of the Membrane

Based on Table 1, the contact angle of the neat PES membrane was 88.01°, while the contact angle of the TiO_2_/PES mixed matrix membrane was 80.53°. The addition of TiO_2_ photocatalysts lowers the water contact angle of the membrane. The water contact angle showed an inverse relationship with hydrophilicity [21]. Therefore, the addition of TiO_2_ photocatalysts made the membrane more hydrophilic. Such a result was due to the presence of a hydroxyl group and an amino group in the TiO_2_ photocatalyst, which caused it to have a high affinity to water and thus improved the hydrophilicity of the membrane [22]. The hydrophilic TiO_2_ photocatalyst reduced the interface energy and moved the TiO_2_ photocatalysts to the membrane’s surface, thus reducing the water contact angle [23].

#### 3.2.4. Membrane Porosity

The results of porosity for a neat PES and the TiO_2_/PES mixed matrix membrane are presented in Table 2. The porosity of the TiO_2_/PES membrane was higher than the neat PES membrane by 5.84%. The porosity of the membrane was affected by the thermodynamics and viscosity of the dope solution [20]. Therefore, the addition of a TiO_2_ photocatalyst made the dope solution viscous, and thus, the membrane became more porous [24].

### 3.3. Membrane Filtration Performance

#### 3.3.1. Effect of Different Concentrations of TiO_2_ Photocatalysts in the Membrane

Figure 6 presents the rejection of humic acid for each of the TiO_2_/PES mixed matrix membrane concentrations. The humic acid rejection of the membrane showed an inverse relationship to the initial pure water flux for different TiO_2_/PES mixed matrix membrane concentrations. The initial pure water flux declined with the increasing TiO_2_/PES mixed matrix membrane concentration. The initial reduction of pure water permeate flux was mainly due to the partial blocking of membrane pores by the TiO_2_ photocatalyst [21]. Such partial blocking made passing water through the membrane pores difficult.

The rejection of humic acid shows an increasing trend with the addition of a TiO_2_ photocatalyst. The rejection of humic acid was improved by plugging the TiO_2_ photocatalyst into membrane pores, thus preventing the humic acid from fouling the membrane pores [21]. Therefore, the increase in TiO_2_/PES mixed matrix membrane concentration is expected to improve the anti-fouling properties.

Figure 7 shows the graph of normalised flux versus time for different TiO_2_/PES mixed matrix membrane concentrations to study the anti-fouling performance and self-cleaning effect. In the first 30 min (section A), the normalised flux for 0 wt.%, 1 wt.%, 3 wt.%, and 5 wt.% of TiO_2_/PES mixed matrix membrane was mostly maintained at 1. This observation was made because the membrane was only passed through with distilled water to determine pure water flux, and thus no fouling occurred.

In the following 120 min (section B), the distilled water was replaced with a humic acid solution, which ran under 10 psi and 250 mL/min. From Figure 8, all the mixed matrices experienced a reduction in normalised flux due to the fouling of humic acid. However, the normalised flux of the mixed/matrix membrane was increased with increasing concentration of TiO_2_ photocatalyst in the membrane, similar to the finding of Wu et al. [22]. A TiO_2_/PES mixed membrane of 0 wt.% had the lowest normalised flux compared to the other mixed matrix membranes.

Hence, a TiO_2_/PES mixed membrane of 0 wt.% had the greatest fouling because of the high hydrophobicity of the PES membrane, which was liable to humic acid deposition. A TiO_2_/PES mixed matrix membrane of 1 wt.%, 3 wt.%, and 5 wt.% had a higher normalised flux than the 0 wt.% TiO_2_/PES mixed matrix membrane. Such an observation was made because the addition of a TiO_2_ photocatalyst improved the hydrophilicity of the membrane, as proved by the result of the contact angle, thus prompting the formation of a hydrophilic water layer that resisted the deposition of humic acid [20,25]. Therefore, the addition of a TiO_2_ photocatalyst enhanced the anti-fouling properties, and the anti-fouling properties can also be further proved by comparing the flux recovery ratio among each TiO_2_/PES mixed matrix membrane concentration.

After 150 min (section C), the humic acid solution was replaced with distilled water to clean the membrane for 10 min to remove loosely bound humic acid on the membrane. Then, the membrane was irradiated with UV light for 20 min for the degradation of the foulant. In Section D, the pure water flux after UV irradiation was tested with distilled water for 10 min. The results for sections C and D for the flux recovery ratio (FRR) at the beginning of the experiment, after the water cleaning process, and after the UV irradiation are shown in Figure 8 for different TiO_2_/PES mixed concentration matrix membranes to evaluate the anti-fouling properties.

Based on Figure 8, the FRR at the beginning of the experiment was mainly the same for a TiO_2_/PES mixed matrix membrane of 0 wt.%, 1 wt.%, 3 wt.%, and 5 wt.%, which are 101.91%, 99.78%, 105.41%, and 99.14%, respectively. For FRR after water cleaning, the FRR for the mixed matrix membrane was higher than a neat PES membrane. The FRR increased from 78% to 110.42% for a TiO_2_/PES mixed matrix membrane, from 0 wt.% to 3 wt.%. The low FRR of a TiO_2_/PES mixed matrix membrane from 0 wt.% to 3 wt.% was due to the neat PES membrane being influenced by the adsorbed humic acid, which cannot be segregated by simple hydraulic washing [13].

Furthermore, the mixed matrix membrane from 0 wt.% to 3 wt.% had better hydrophilicity by adding TiO_2,_ which enhanced the flux recovery [13]. The result from ATR-FTIR analysis proved the hypothesis, which consists of the -OH group at 3436 cm^−1^. Therefore, a TiO_2_/PES mixed matrix membrane of 3 wt.% had the best anti-fouling properties because it was easy to clean compared to the other mixed matrix membranes. After that, the FRR was reduced to 94.44% for the TiO_2_/PES of 5 wt.% mixed matrix membrane. A high concentration of TiO_2_ photocatalyst blocked the membrane pores and obstructed the interaction of PES and solvent molecules [21].

After UV irradiation for 20 min, the FRR gradually increased from 80.42% to 111.83% for the TiO_2_/PES mixed matrix membrane of 0 wt.% to the TiO_2_/PES mixed matrix membrane of 3 wt.% and then dropped to 108.2% for a TiO_2_/PES mixed matrix membrane of 5 wt.%. The FRR value after UV irradiation of a TiO_2_/PES mixed matrix membrane of 0 wt.% was almost similar to the FRR value after water cleaning because there was an absence of a TiO_2_ photocatalyst to perform the degradation of humic acid. The high FRR for the mixed matrix membrane was mainly due to the TiO_2_ photocatalyst, a semi-conductor that can be activated by UV-irradiation with rays equal to or greater than the band gap energy, which leads to the transfer of an electron from the capacity band to the conduction band. The oxygen molecules in the environment reacted with the photo-generated electrons and formed superoxide radical anions (O_2_^−^).

Conversely, the photo-generated holes reacted with the water molecules in the environment and formed OH radicals. The superoxide radical anions and OH radicals were strong oxidant reagents that could decompose the foulant deposited on the membrane [21]. Therefore, a TiO_2_/PES mixed matrix membrane of 3 wt.% showed the best self-cleaning effect in decomposing humic acid.

#### 3.3.2. Effect of Duration of UV Irradiation

A TiO_2_/PES mixed matrix membrane of 3 wt.% had the best anti-fouling performance, so it was used to determine the effective duration for UV irradiation. Figure 9 demonstrates the flux recovery ratio (FRR) graph at the beginning of the experiment and after UV irradiation versus the duration of UV irradiation.

From Figure 9, the FRR after UV irradiation showed an increasing trend with longer UV irradiation duration. A neat PES membrane had the lowest FRR after irradiation of 81.23%, because of the absence of a TiO_2_ photocatalyst. For a TiO_2_/PES mixed matrix membrane of 3 wt.%, the FRR after irradiation increased with the duration of UV irradiation (from 2 to 20 min) from 93.18% to 111.18%. The FRR after irradiation for 2 and 10 min of UV irradiation was 93.18% and 95.64%, respectively. The observed trend was due to insufficient energy to transfer the electron to the conduction band. The highest FRR after irradiation of a TiO_2_/PES mixed matrix membrane of 3 wt.% was during the first 20 min of UV irradiation. These results showed that 20 min of UV irradiation was enough to transfer an electron from the capacity band to the conduction band and thus produce superoxide radical anions (O_2_^−^) and OH radicals for photocatalytic degradation of humic acid deposited on the membrane [21].

### 3.4. Membrane Fouling Behaviour Models

The membrane’s fouling behaviour in a cross-flow filtration system can be determined using the most widely recognised model, Hermia’s model. Some researchers have also applied Hermia’s models to analyse membrane fouling behaviour. Rayess et al. [9] reported using Hermia’s models for analysing membrane fouling in cross-flow microfiltration of wine, while Kazemi et al. [26] reported using the same models for cross-flow microfiltration of ceramic membrane fouling. Thus, it can be concluded that these mathematical models are considered popular in detecting the fouling mechanism of membranes. An important assumption is made by applying Hermia’s models to a cross-flow filtration system. It is assumed that the flow in the cross-flow filtration system is in a quasi-steady-state condition as the permeate flux is almost constant with time for a prolonged duration and reduces gradually [27]. According to Iritani and Katagiri [28], there are four different fouling behaviours regarding the deposition of particles on the membranes: the complete blocking model, the standard blocking model, the intermediate blocking model, and the cake filtration model. The simplified general fouling model’s equation developed by Hermia is shown in Equation (3) [26].
(3)$$\frac{d^{2}t}{dV^{2}}=K{\left(\frac{dt}{dV}\right)}^{n}$$
where *t* is the filtration duration, *V* is the cumulated filtrate volume, *K* is the proportionality constant, and *n* is a constant represented by different numbers according to other models. Based on Equation (3), the linearised equation derived for all the models at continuous pressure filtration, along with their graphical approaches, schematic diagrams, and model descriptions, is presented in detail in Table 3. These membrane fouling models can be used individually or combined to explain membrane fouling behaviour [29]. Table 3 shows the membrane fouling model: cake filtration, standard blocking, intermediate blocking, and complete pore blocking.

In this study, a TiO_2_/PES mixed matrix membrane of 3 wt.% was used to investigate the fouling model of a TiO_2_/PES mixed matrix membrane. The result from permeate flux was used to plot a graph representing four different fouling models, and the plotted graph was compared with the theoretical model by indicating the R^2^ value as shown in Figure 10a–d. Based on Figure 10d, the intermediate blocking model showed the highest R^2^ value, which was 0.986, followed by the complete blocking fouling model (Figure 10a with an R^2^ equal to 0.9687). The cake filtration model (Figure 10b) had an R^2^ value of 0.514, while the standard blocking model (Figure 10c) had the lowest R^2^ value of 0.4711. Therefore, the intermediate blocking model is the best prediction model, as the R^2^ value from a linearised graph was nearest to 1.

In this case, the intermediate pore-blocking mechanism best describes the fouling behaviour of a TiO_2_/PES mixed matrix membrane of 3 wt%. However, it is interesting that only the complete and intermediate pore-blocking models yielded a higher R^2^ value than the others. Such a finding is mainly because both complete and intermediate pore-blocking models were occasionally known as the precursors of cake filtration and were responsible for the initial stage of pore-blocking [15]. In this study, the intermediate pore-blocking mechanism can explain the membrane more appropriately than the complete pore-blocking mechanism, as the membrane did not separate the permeate, which can be proven by the permeate rejection of roughly 89.9%, as shown in Figure 6. This result makes the intermediate pore-blocking mechanism a better predictor of the membrane’s fouling behaviour [30].

Intermediate pore blocking is often associated with the particles being deposited or adsorbed on the surface of the membrane or the particles themselves [8]. Generally, this phenomenon occurs whenever the particle’s size is similar to the membrane pore size [31]. However, this mechanism seems to only happen at the early stages of membrane fouling and will eventually lead to cake filtration [32]. The result specifies that a TiO_2_/PES mixed matrix membrane of 3 wt% has most likely prolonged the phase for intermediate pore blocking. The benefit of a POM/PES mixed matrix membrane of 3 wt% is that it can maintain the high permeate flux for a long time during the intermediate pore-blocking phase, as once the cake is formed, the permeate flux would be much lower. Moreover, the intermediate pore blocking due to the accumulated foulants on the top surface of the membrane allows the membrane pores to have reversible fouling [33].

## 4. Conclusions

A TiO_2_/PES mixed matrix membrane of 3 wt% was observed to have the best anti-fouling and self-cleaning effect with improved hydrophilicity. The optimum duration for UV irradiation of a TiO_2_/PES mixed matrix membrane was 20 min. Furthermore, the fouling behaviour of a mixed matrix membrane was fitted to the intermediate blocking model. The prolonged intermediate pore blocking and its later phase, cake filtration, are said to be beneficial to membrane fouling as most of the fouling that occurs would be reversible. This indicates that the TiO_2_/PES mixed matrix membrane is capable of providing a longer membrane lifespan. Furthermore, adding TiO_2_ to the PES membrane enhances its anti-fouling and self-cleaning properties.

## Figures and Tables

**Figure 1 membranes-13-00373-f001:**
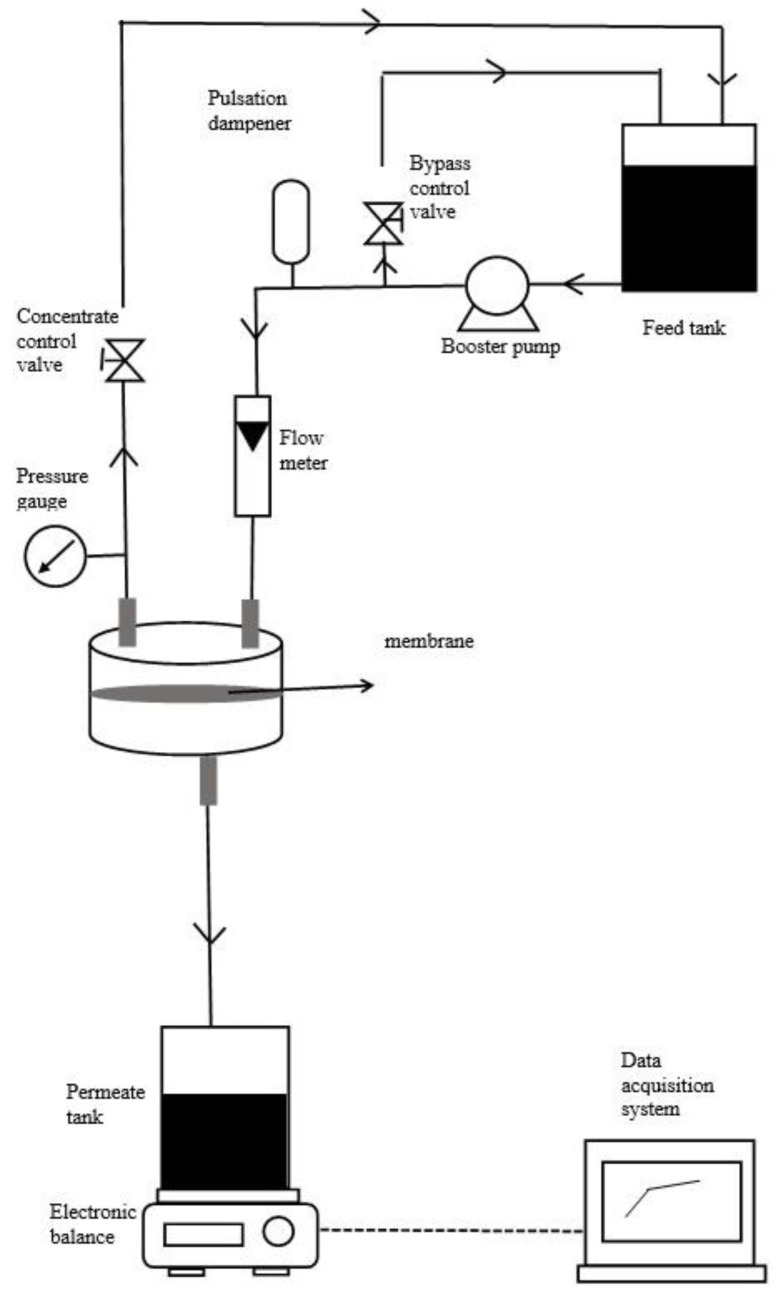
Schematic diagram of a cross-flow filtration system.

**Figure 2 membranes-13-00373-f002:**
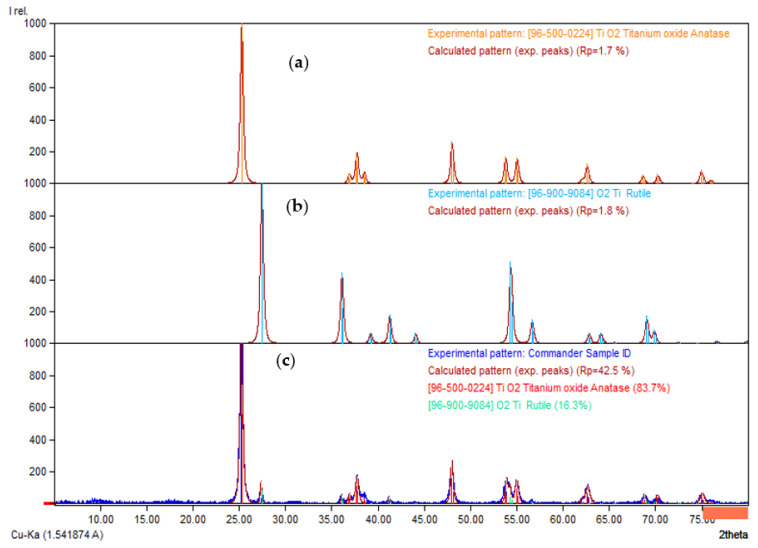
XRD patterns for TiO_2_ photocatalysts (**a**) TiO_2_ anatase-phase database. (**b**) TiO_2_ rutile phase database. (**c**) XRD experimental pattern of TiO_2_ photocatalyst.

**Figure 3 membranes-13-00373-f003:**
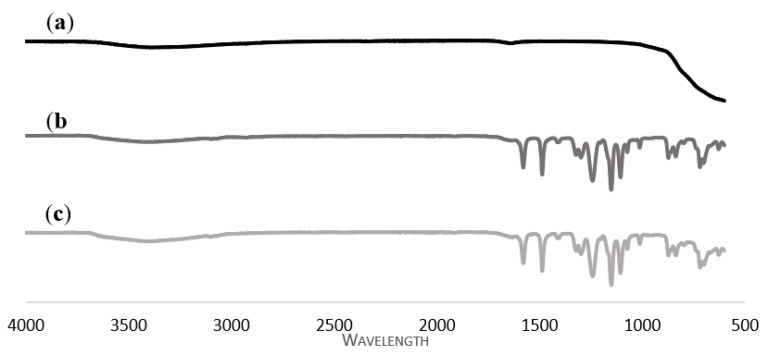
ATR-FTIR spectra of (**a**) TiO_2_ photocatalyst; (**b**) a neat PES membrane; and (**c**) a TiO_2_/PES mixed matrix membrane.

**Figure 4 membranes-13-00373-f004:**
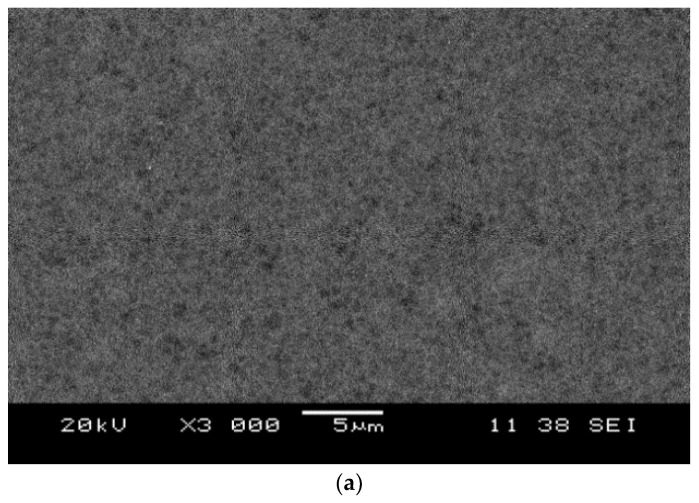
(**a**) Top view SEM images of a neat PES membrane; and (**b**) top view SEM images of the TiO_2_/PES mixed matrix membrane.

**Figure 5 membranes-13-00373-f005:**
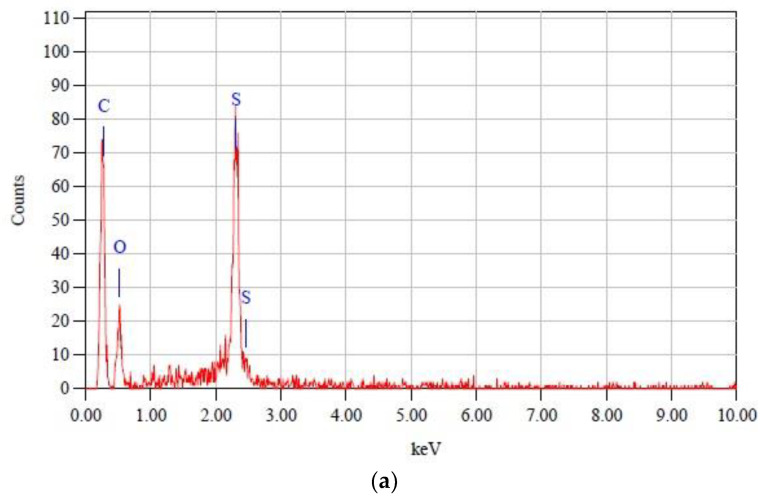
(**a**) EDX spectrum of a neat PES membrane; and (**b**) EDX spectrum of the TiO_2_/PES mixed matrix membrane.

**Figure 6 membranes-13-00373-f006:**
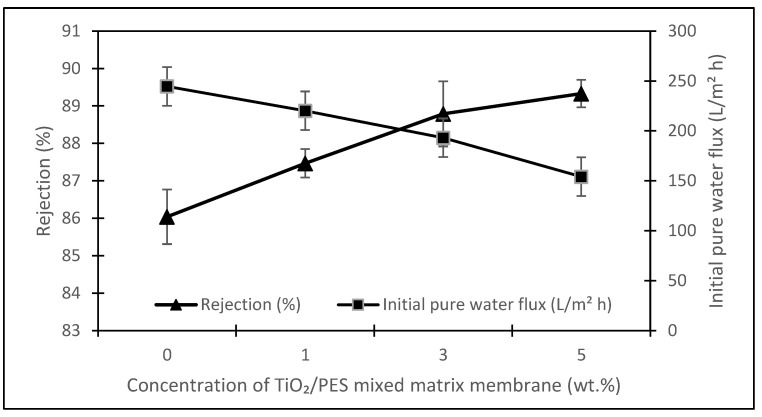
Humic acid rejection and initial pure water flux for each TiO_2_/PES mixed matrix membrane concentration.

**Figure 7 membranes-13-00373-f007:**
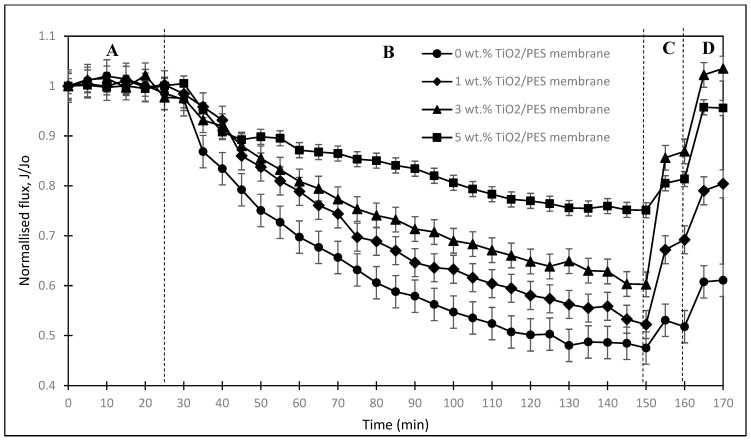
Graph of normalised flux versus time for different TiO_2_/PES mixed matrix membrane concentrations. (Region A: Feed with DI water to allow the membrane to reach quasi-equilibroum steady state; Region B: HA filtration process; Region C: membrane cleaning process (DI water); Region D: Membrane cleaning process (UV irradiation plus water cleaning)).

**Figure 8 membranes-13-00373-f008:**
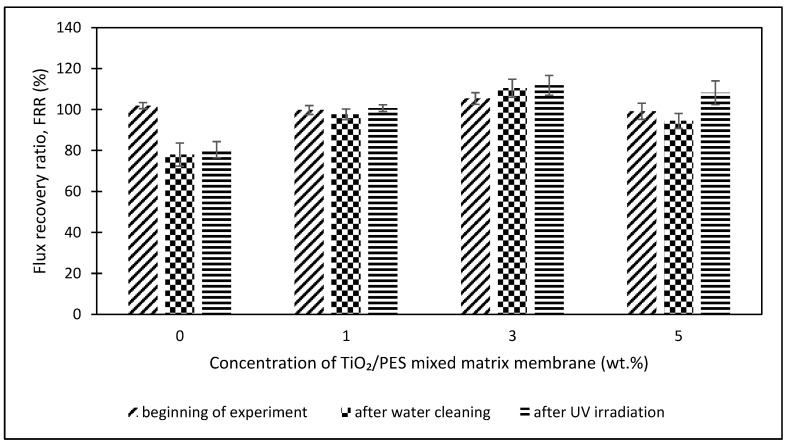
Comparison of FRR among different concentrations of TiO_2_/PES mixed matrix membrane at the beginning of the experiment, after water cleaning, and after UV irradiation.

**Figure 9 membranes-13-00373-f009:**
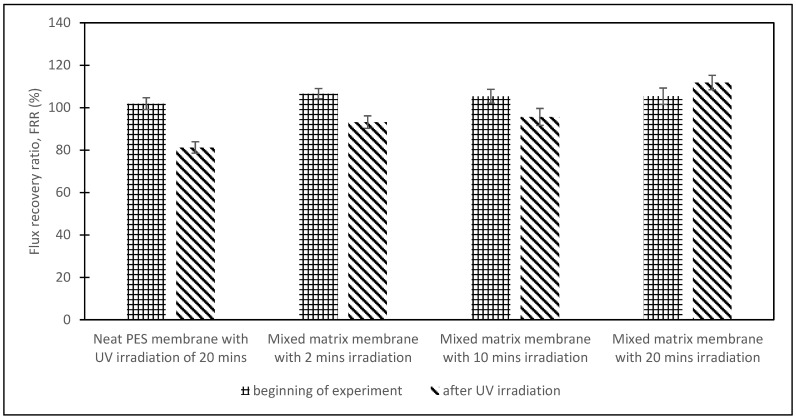
Comparison of FRR among different durations of UV irradiation at the beginning of the experiment and after UV irradiation for a TiO_2_/PES mixed matrix membrane of 3 wt.%.

**Figure 10 membranes-13-00373-f010:**
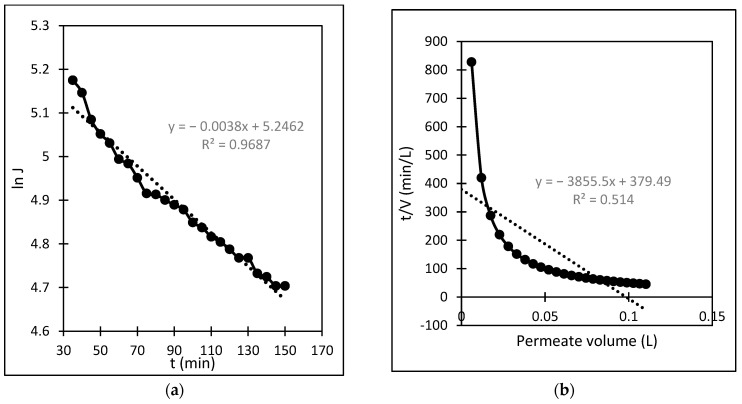
(**a**). Linearised graph representing complete blocking; (**b**) linearised graph representing cake filtration; (**c**) linearised graph representing standard blocking; and (**d**) linearised graph representing intermediate blocking.

**Table 1 membranes-13-00373-t001:** Contact angle measurement for neat a PES membrane and the TiO_2_/PES mixed matrix membrane.

Type of Membrane	Contact Angle (°)
Neat PES membrane	88.01 ± 0.77
TiO_2_/PES mixed matrix membrane	80.5 ± 30.89

**Table 2 membranes-13-00373-t002:** The porosity of a neat PES membrane and the TiO_2_/PES mixed matrix membrane.

Membrane	Porosity (%)
Neat PES membrane	80.5 ± 0.81
TiO_2_/PES mixed matrix membrane	86.34 ± 0.87

**Table 3 membranes-13-00373-t003:** Hermia’s models for constant pressure filtration [26,27].

Model	Linearized Equation	Graphical Approaches	Schematic Diagram	Description
Cake filtration (n = 0)	$\frac{t}{V}=\frac{K_{c}}{2}V+\frac{1}{J_{0}}$	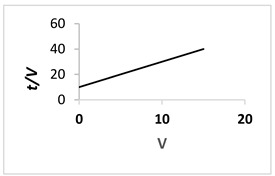	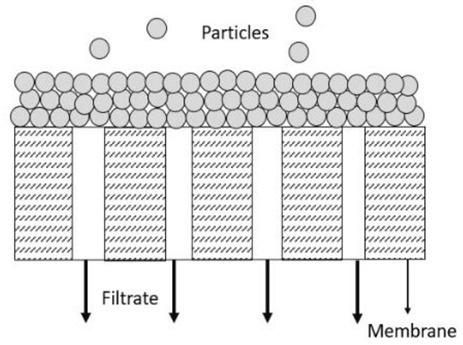	The size of particles is larger than the pore size of membrane and unable to pass though the pores, forming the cake layer on the membrane surface.
Intermediate blocking (n = 1)	$\frac{1}{J}=K_{i}t+\frac{1}{J_{0}}$	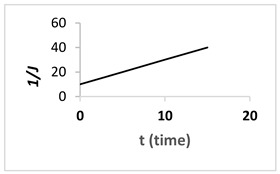	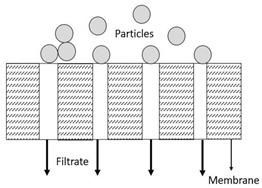	The fouling happens as the particle is having the similar size towards the pore size of the membrane.
Standard blocking (n = 1.5)	$\frac{t}{V}=\frac{K_{s}}{2}t+\frac{1}{J_{0}}$	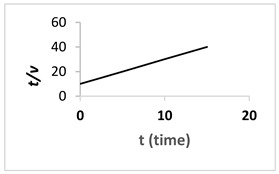	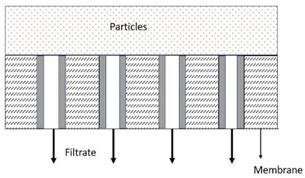	The fouling occurs due to the smaller particles than the pores size of membrane are deposited within the membrane pore.
Complete blocking (n = 2)	$V=\frac{J_{0}}{K_{b}}\left(1-e^{-K_{b}J}\right)$	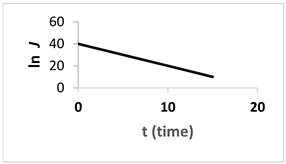	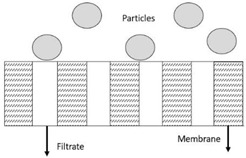	The inlet pores of the membrane completely blocked by all particles on the membrane surface.

## Data Availability

Not applicable.

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
