# Peer review of "Synthesis and Characterisation of Self-Cleaning TiO2/PES Mixed Matrix Membranes in the Removal of Humic Acid"

_membranes, 2023, doi:10.3390/membranes13040373_

Round 1

Reviewer 1 Report

The manuscript entitled “Synthesis and Characterisation of Self-cleaning TiO2/PES Mixed Matrix Membrane in Removal of Humic Acid” is an interesting work on the evaluation of a photocatalytic membrane towards pollutant removal. Some specific comments are below:

1) Keywords need to be improved. Besides “membrane fouling” all the other words are already present in the title.

2) Lines 41-43. Authors should describe the 2 modes of using TiO2 and membranes (i.e. with the catalyst in suspension or within the membrane matrix), and then explain the advantages of using it in the mixed membrane matrix. Please check: 10.1016/j.cej.2019.122114.

3) Line 45: Surprisingly, the photocatalytic activity of the membranes was not deeply evaluated in this work. Authors should at least describe this phenomenon. PES/TiO2 membranes have been widely evaluated for the photocatalytic degradation of pollutants. Please check: 10.3390/catal10070725.

4) In topic 2.1, authors must give more information about the humic acid used and the TiO2 (for example, crystallization).

5) In topic 3.3.2, why the photocatalytic degradation of the humic acids was not discussed? The author explained this topic only based on the FRR. It would be interesting to see a graph of the membrane flux and humic acid profile within the time.

Reviewer 2 Report

Please explain the detail of how to get apparent volume and existence volume

The author need to mention on how to measure the concentration of humic acid in the water

Please add cross section images of the membranes for every concentration of TiO2/PES

Please add error bar for FIg. 7,8 9, 10

Reviewer 3 Report

In this work, the author blended TiO2 into PES membrane matrix and evaluated its self-cleaning antifouling behavior for humic acid. The results and discussion are reasonable. After careful evaluation, I think it must perform a major revision.

1. The main weakness of this work is the poor novelty. It is unclear why the author especially embedded TiO2 nanoparticles into PES membrane. Similar research works (e.g., Applied Surface Science, 255(9), 4725-4732; Journal of Membrane Science, 467, 226-235; Desalination, 292, 19-29) have been widely reported by others. What’s the main contribution of this work??

2. The logic of this work should be improved. In my opinion, the blending concentration of TiO2 nanoparticles into PES membrane matrix could be optimized first and then characterizes the synthesized membrane under the best conditions (e.g., SEM, ATR-FTIR, and contact angle). Otherwise, the characterization must be done for all synthesized membranes under different blending concentrations (i.e., 0-5 wt.%).

3. The standard deviation must be provided (e.g., Tables 1-2, Figures 7-10). Captions for all tables and Figures (1,3) can be made more descriptive. The resolution of some figures should be improved. 

4. It is well-known that the PES membrane is not stable under UV light (Journal of Colloid and Interface Science 517 (2018): 155-165). Did the author check the changes in membrane separation behaviors (i.e., rejection and permeability) after 2-20 min UV irritation? Some characterizations (e.g., SEM) can be performed to show detailed changes (or no changes).

5. The membrane fouling phenomenon is directly related to membrane flux. To prove membrane antifouling activity, the HA filtration experiment must also be performed at a similar initial membrane flux (i.e., different TMP but the same initial flux). The author can read and follow some reported works. For example, Journal of Colloid and Interface Science 613 (2022): 426-434; Journal of Colloid and Interface Science 517 (2018): 155-165;

Round 2

Reviewer 1 Report

The authors have improved the manuscript accordingly.

Author Response

No further comments from reviewer 1. Thank you.

Reviewer 2 Report

The manuscript can be accepted as it is

Author Response

No further comments from Reviewer 2. Thank you.

Reviewer 3 Report

The authors didn’t carefully address all of my comments. As a researcher and responsible reviewer, I can’t accept this version.

1. The authors only did the characterizations (e.g., XRD, FTIR, and SEM) for the 3 wt.% of TiO2 blended membrane. Why? How the authors proved that this is the best condition without the condition optimization first? 

2. The resolution for each figure must be improved (e.g., Figures 1-5 and Table 3). Could the authors read it? Where is the standard deviation in Tables 1-2? This is a scientific paper, not a research report. The authors should know that lots of readers will read it later.

3. Again, the fouling experiment must also be performed at a similar initial flux (membrane flux is different, see Figure 6). Without this data, the authors can’t claim that the synthesized membrane is better than the unmodified membrane. Normalized flux in Figure 7 doesn’t mean a similar membrane filtration flux. 

Round 3

Reviewer 3 Report

Good job!